# MLH1 Promoter Variant −93G>A and Breast Cancer Susceptibility: Evidence from Azerbaijan

**DOI:** 10.3390/biomedicines13112769

**Published:** 2025-11-12

**Authors:** Nigar Karimova, Bayram Bayramov, Zumrud Safarzade, Nigar Mehdiyeva, Hagigat Valiyeva

**Affiliations:** 1Laboratory of Human Genetics, Genetic Resources Institute, Ministry of Science and Education, Baku AZ1106, Azerbaijan; zum.mukh@gmail.com; 2Department of Oncology, Oncological Clinic, Azerbaijan Medical University, Baku AZ1022, Azerbaijan; ladymed77@hotmail.com (N.M.); doctorhagigat1@gmail.com (H.V.)

**Keywords:** breast cancer, MMR genes, PCR-RFLP, *MLH1* −93G>A

## Abstract

**Background:** Breast cancer (BC) is the most common malignancy among women, and genetic predisposition plays a critical role in its development. Among DNA mismatch repair (MMR) genes, *MLH1* is essential for maintaining genomic stability, and promoter variants may influence its transcriptional regulation. Variants in MMR genes, including *MLH1*, have been implicated in cancer susceptibility; however, evidence regarding the promoter polymorphism −93G>A (rs1800734) and its association with BC remains limited and inconsistent across populations. **Methods**: We conducted a case–control study of 143 breast cancer patients and 161 cancer-free controls of Azerbaijani origin. Genotyping of *MLH1* −93G>A was performed using PCR-RFLP and validated by next-generation sequencing (NGS). Odds ratios (ORs) and 95% confidence intervals (CIs) were calculated under different genetic models by logistic regression, followed by false discovery rate (FDR) correction for multiple testing. **Results**: The genotype distribution among patients was 25.9% GG, 58.7% GA, and 15.4% AA, compared with 37.9%, 46.6%, and 15.5% in controls. A significant association was observed between the GA genotype and BC risk (OR = 1.855, 95% CI: 1.104–3.085, *p* = 0.019). In the dominant model (GA + AA vs. GG), carriers of the A allele showed increased breast cancer risk (OR = 1.747, 95% CI: 1.069–2.856, *p* = 0.026). Genotype distribution was also associated with tumor grade (*p* = 0.047) and stage (*p* = 0.013). However, none of the associations remained significant after FDR adjustment. **Conclusions**: This pilot study provides the first evidence from Azerbaijan suggesting a potential role of the *MLH1* −93G>A variant in breast cancer susceptibility. Although the associations were nominal and require validation in larger cohorts, the findings point to a biologically plausible link between *MLH1* promoter variation and impaired MMR activity, which may contribute to polygenic breast cancer risk. These preliminary results emphasize the importance of evaluating MMR gene variants in underrepresented populations and support further studies integrating functional assays and broader gene coverage.

## 1. Introduction

It is common knowledge that breast cancer (BC) is the most frequently diagnosed malignancy among women worldwide, remaining a major public health challenge. In 2022, the disease accounted for approximately 2.3 million new cases and 670,000 deaths, representing nearly one quarter of all new female cancer cases. If current trends persist, the global burden is estimated to rise by nearly 38% in incidence and 68% in mortality by 2050, with the heaviest consequences falling on women in low- and middle-income countries. Recent global estimates suggest that one in twenty women will develop breast cancer during their lifetime, and one in seventy will die from the disease [1].

The development of BC is shaped by a combination of environmental, hormonal, and genetic factors. While lifestyle factors such as reproductive history, diet, and physical activity shape BC risk, inherited susceptibility often determines who eventually develops the disease and at what age. Alterations in high-penetrance genes like BRCA1 and BRCA2 explain a proportion of familial breast cancers; however, they leave most sporadic cases unexplained [2]. This gap has shifted attention toward more common, low-penetrance variants, notably single-nucleotide polymorphisms (SNPs), which may influence risk in more subtle ways [3].

Among candidate pathways, the DNA mismatch repair (MMR) system is crucial for correcting replication errors and protecting genomic integrity. Defects in the mismatch repair system cause microsatellite instability and an increased mutation load, which are hallmarks observed in colorectal and endometrial cancers, and in a subset of breast tumors [4]. The human MutL homolog 1 (*MLH1*) gene is central to MMR activity, and its disruption through germline mutations or promoter hypermethylation is well documented [5,6].

The *MLH1* −93G>A variant (rs1800734) has attracted considerable attention because of its functional consequences and broad clinical associations. Located in the promoter region, this polymorphism may alter transcription factor binding, modify CpG island methylation, and ultimately reduce MLH1 expression [3,5]. Several studies have linked the A allele to promoter hypermethylation and microsatellite instability, particularly in sporadic colorectal cancer [7,8]. Associations have also been described with gastric, lung, colorectal and endometrial cancers, although the strength and consistency of these findings vary across populations [9,10,11,12,13].

In BC, evidence on *MLH1* polymorphisms and mismatch repair deficiency remains limited. Wen et al. reported *MLH1* promoter methylation and the loss of MMR protein expression in a subset of breast tumors, suggesting that this pathway may play a role in tumorigenesis [5]. Population-based analyses have also indicated that the impact of *MLH1* variants may vary across ethnic groups [14]. While a Korean case–control study suggested a modest association between *MLH1* polymorphisms and BC risk, subsequent studies in other populations, including Chinese and Iranian cohorts, did not confirm significant effects [15,16,17,18]. These inconsistencies underscore the need for further investigations across diverse populations.

Genetic studies on BC in Azerbaijani women remain scarce, and, to date, there is no study investigating the role of *MLH1* promoter polymorphisms in this disease. While *MLH1* variants have been studied in colorectal cancer within Azerbaijan [19], their relevance to breast cancer has not been addressed. Given the unique genetic background of the South Caucasus region and the absence of breast cancer–specific data, investigating the *MLH1* −93G>A variant in our cohort may provide valuable insights into breast cancer susceptibility and help fill an important gap in regional cancer genetics.

To our knowledge, this is the first study from Azerbaijan to investigate the *MLH1* −93G>A polymorphism in breast cancer patients, offering new evidence on genetic risk within this underrepresented population and contributing to the broader understanding of MMR gene variants in carcinogenesis.

## 2. Materials and Methods

### 2.1. Study Subjects

This case–control study included 143 women with histologically confirmed breast cancer and 161 cancer-free controls. Recruitment was carried out between 2022 and 2024 at the Oncology Clinic of the Azerbaijan Medical University. Patients were newly diagnosed and had not received chemotherapy or radiotherapy before blood sampling.

Diagnosis was confirmed histopathologically, and only sporadic cases were included; individuals with a family history of breast or ovarian cancer were excluded. Among the tested cases, BRCA1/2 status was determined for most patients, with the majority being BRCA-negative. Clinical and pathological data—including age, tumor grade and stage, histological type, and hormone receptor status (ER, PR, HER2)— were obtained from official hospital pathology and clinical records provided by the Oncology Clinic of the Azerbaijan Medical University. Height and weight were measured at diagnosis to calculate BMI (kg/m^2^).

Controls were age-matched women attending the same clinic for routine health check-ups. None had a history of breast or other cancers. To ensure comparability and reduce confounding, individuals with acute or chronic illnesses, active infections, hepatitis B or C, HIV infection, or other systemic diseases were excluded. Both patients and controls were of Azerbaijani origin to avoid population stratification (Table 1).

All participants provided written informed consent before enrollment. The study protocol was approved by the Ethics Committee of the Ministry of Science and Education of the Republic of Azerbaijan, Genetic Resources Institute, and was conducted according to the Declaration of Helsinki.

Peripheral blood (2 mL) was collected in EDTA tubes (Greiner Bio-One GmbH, Kremsmünster, Austria) from each subject. Genomic DNA was extracted from leukocytes using the salting-out method as described by Miller et al. [20]. DNA concentration and purity were assessed spectrophotometrically with a NanoDrop 2000 (Thermo Scientific, Waltham, MA, USA), and samples were stored at −20 °C until analysis.

### 2.2. Genotyping

The *MLH1* −93G˃A (rs1800734) polymorphism was analyzed using the PCR-RFLP method. PCR reactions (20 µL total volume) were performed using a T100 Thermal Cycler (Bio-Rad Laboratories, Hercules, CA, USA) with reagents from Thermo Fisher Scientific (Waltham, MA, USA) following the standard protocol. PCR conditions involved an initial denaturation at 95 °C for 5 min, followed by 30 cycles comprising denaturation at 95 °C for 30 s, annealing at 58 °C for 1 min, and extension at 72 °C for 2 min, with a final elongation at 72 °C for 5 min. The primer sequences used for the genotyping of the *MLH1* −93G˃A (rs1800734) polymorphism were as follows: Forward primer: 5′-CCGAGCTCCTAAAAACGAAC-3′, reverse primer: 5′-CTGGCCGCTGGATAACTTC-3′.

During the PCR-RFLP analysis, the 387 bp amplified fragment of the target gene was enzymatically digested using PvuII (New England Biolabs, Ipswich, MA, USA) at 37 °C for 5 h. The resulting fragments were separated by electrophoresis on a 2% agarose gel (Sigma-Aldrich, St. Louis, MO, USA) using a PowerPac Basic Electrophoresis System (Bio-Rad Laboratories, Hercules, CA, USA). The digestion patterns corresponded to the following genotypes: the GG homozygous wild type genotype remained undigested, producing a single 387 bp fragment; the AA homozygous genotype was completely cleaved, yielding two fragments of 207 bp and 180 bp; and the GA heterozygous genotype exhibited three distinct bands at 387 bp, 207 bp, and 180 bp. To confirm the results, 10% of the randomly selected samples were retested using the same PCR-RFLP method, and the findings were fully consistent.

### 2.3. Quality Control

Genotype distributions in the control group were examined for Hardy–Weinberg equilibrium using the chi-square test. Samples with unclear results or poor DNA quality were excluded.

For additional validation, 10% of randomly selected samples were reanalyzed by next-generation sequencing (NGS) (Illumina MiSeq, San Diego, CA, USA). The NGS results were in full agreement with the PCR-RFLP data, validating the accuracy of genotyping.

### 2.4. Statistical Analysis

Genotype and allele frequencies were compared between cases and controls using Pearson’s chi-square test (χ^2^) or Fisher’s exact test, where appropriate. For contingency tables larger than 2 × 2 with small counts, Fisher’s exact test was performed using the Social Science Statistics calculator (http://www.socscistatistics.com/tests/chisquare2/Default2.aspx; accessed on 10 March 2025).

Odds ratios (ORs) with 95% confidence intervals (CIs) were estimated using binary logistic regression to evaluate associations under dominant, recessive, and allelic models. Statistical analyses were performed with SPSS software (version 26.0; IBM Corp., Armonk, NY, USA).

All *p*-values were two-sided, with *p* < 0.05 considered statistically significant. Multiple comparisons were adjusted using the Benjamini–Hochberg false discovery rate method.

## 3. Results

This study was conducted on 143 patients diagnosed with breast cancer and 161 cancer-free controls. The patients’ characteristics are presented in Table 1. The age range of the patients was 39–85 years, while in the control group, it was 32–83 years. The mean age was 62 ± 11 years in patients and 60 ± 11 years in controls, with no significant difference (*p* = 0.432). The mean BMI of patients was 28.1 ± 4.9 kg/m^2^, corresponding to the overweight range, whereas controls had a mean BMI of 23.5 ± 3.8 kg/m^2^ (*p* < 0.001).

Histologically, invasive ductal carcinoma predominated (85.3%), followed by invasive lobular carcinoma (12.6%) and other special types (2.1%). Hormone receptor analysis showed ER positivity in 67.1% and PR positivity in 63.6% of cases, while HER2 overexpression was observed in 19.5%. Triple-negative tumors represented 4.9% of the cohort.

Among patients, the majority were diagnosed with grade II tumors (72.0%), followed by grade I (18.2%) and grade III (9.8%). Tumor stage distribution was 9.8% stage I, 41.2% stage II, 9.8% stage III, and 39.2% stage IV.

Serum CA 15-3 levels ranged from 26 to 300 U/mL (mean ± SD = 78.4 ± 52.6). Among the tested cases, BRCA1/2 status was available for most patients, with 79 (55%) testing negative.

Genotype frequencies of the *MLH1* −93G>A polymorphism in patients and controls are shown in Table 2. In the control group, genotype distributions were consistent with the Hardy–Weinberg equilibrium (χ^2^ = 0.060, *p* = 0.807), supporting the reliability of genotyping.

Among the patients, the distribution of genotypic frequencies was as follows: 25.9% GG, 58.7% GA, and 15.4% AA. Similarly, in the control group, the frequencies were 37.9% GG, 46.6% GA, and 15.5% AA. The GA genotype was associated with an increased susceptibility to breast cancer (OR = 1.855, 95% CI: 1.104–3.085, *p* = 0.019).

In the dominant model (GA + AA vs. GG), a significant association was detected (OR = 1.747, 95% CI: 1.069–2.856, *p* = 0.026). The recessive model (AA vs. GG + GA) did not show significance (OR = 0.989, 95% CI: 0.530–1.844, *p* = 0.972).

In the allelic comparison, the A allele frequency was higher in patients (44.8%) than in controls (38.8%), corresponding to an OR of 1.277 (95% CI: 0.924–1.764, *p* = 0.138).

A post hoc power analysis indicated that, with 143 cases and 161 controls, the study had approximately 67% power to detect an effect size at a significance level of 0.05.

These associations are also visualized in Figure 1, which illustrates the relative effect sizes for each genetic model.

Table 3 presents the distribution of genotypes based on the age-stratified analysis. Among participants aged ≤ 60 years, the genotype frequencies in patients were 26.9% for GG, 62.7% for GA, and 10.4% for AA, compared with 38.4%, 45.8%, and 15.8% in controls, respectively. The GA genotype was more frequent among patients, whereas the AA genotype was slightly more common in controls. Logistic regression indicated that GA carriers had a borderline increased risk compared with GG (OR = 1.948, 95% CI: 0.976–3.891, *p* = 0.050), while no association was observed for the AA genotype (*p* > 0.05).

In participants aged > 60 years, patients had genotype frequencies of 25% GG, 55.2% GA, and 19.8% AA, compared with 37%, 48.2%, and 14.8% in controls. Both GA and AA genotypes were more common among patients; however, neither comparison reached statistical significance (*p* > 0.05).

The distribution of genotype frequencies in tumor grade and stage is presented in Table 4. For tumor grade, the GA genotype was most frequent among grade II cases (62.2%), whereas the AA genotype was more common in grade III (35.7%). Overall, a significant association was observed between genotype distribution and tumor grade (*p* = 0.047).

For tumor stage, GA carriers were particularly frequent in stage IV tumors (69.5%), while AA genotypes appeared more often in stage I (35.7%). Genotype distribution differed significantly across stages (*p* = 0.013).

No significant differences in *MLH1* −93G>A genotype frequencies were observed across histological subtypes (*p* = 0.76; Appendix A).

## 4. Discussion

Common genetic variants in DNA repair pathways represent an important but still incompletely defined layer of cancer susceptibility. While high-penetrance mutations in BRCA1/2 account for a fraction of hereditary breast cancers, most disease risk arises from the cumulative effects of more frequent, low-penetrance alleles [21,22]. Within the MMR pathway, the *MLH1* −93G>A (rs1800734) promoter variant has drawn attention for its potential to alter transcription factor binding and CpG methylation, leading to reduced MLH1 expression [3,6,23]. Reduced MLH1 activity compromises DNA repair fidelity and is a well-recognized driver of microsatellite instability (MSI) in colorectal and endometrial cancers [4,6,9,13]. Although biologically plausible, the relationship between this variant and breast cancer has remained inconsistent across populations.

In our cohort, carriers of the A allele (GA + AA) showed a higher risk of breast cancer under the dominant model. The effect size was modest, in line with what would be expected for common variants. Importantly, this signal did not persist after Benjamini–Hochberg correction, indicating that the association should be interpreted cautiously. Nevertheless, the pattern is biologically consistent and resonates with prior studies suggesting that reduced MLH1 activity contributes to carcinogenesis, although direct evidence in breast cancer remains mixed. For example, a Korean case–control study reported a higher risk associated with the GG genotype, while a broader Asian meta-analysis found no significant link between this polymorphism and breast cancer, though an increased risk was noted for lung cancer under the recessive model [15,16]. In contrast to our results, Kappil et al. investigated twelve polymorphisms across key MMR genes, including *MLH1* rs1800734, and found no significant association with disease susceptibility [24].

No studies to date have specifically evaluated the association of *MLH1* −93G>A with breast cancer risk in European populations. Available European data are mostly derived from colorectal and endometrial cancers, where rs1800734 shows inconsistent effects, indicating potential tissue-specificity [13]. Collectively, these findings highlight the variability of results and underscore the importance of conducting larger, ethnically diverse studies.

Evidence from other malignancies supports the functional plausibility of our findings. In colorectal cancer, the −93G>A variant is consistently linked to microsatellite instability and elevated risk [9,12]. Similar promoter-dependent downregulation of MLH1 has been observed in endometrial carcinoma, where hypermethylation predicts poorer prognosis [25,26]. These observations suggest that impaired mismatch repair caused by *MLH1* promoter variation may act as a shared susceptibility mechanism across tumor types, even if its magnitude differs by tissue context.

Functional studies demonstrate that −93G>A alters transcription factor binding and increases promoter methylation, resulting in reduced MLH1 expression [3]. Within Azerbaijan, Bayramov et al. reported that MMR polymorphisms contribute to colorectal carcinogenesis, reinforcing the significance of this pathway in local populations [19]. Importantly, *MLH1* promoter methylation and reduced expression have also been associated with more aggressive phenotypes across multiple cancers, including breast, endometrial, and colorectal malignancies [25,26].

At a broader level, meta-analyses of MMR variants indicate that *MLH1* polymorphisms can modestly influence breast cancer susceptibility, though the specific contribution of rs1800734 remains uncertain. For example, Zhang et al. reported significant associations for rs1799977 and rs63750447, particularly in Caucasians, but not for rs1800734 [27]. Studies from Iranian cohorts likewise found no effect for promoter variants rs63749795 and rs63749820 [17,18]. Meanwhile, large-scale genomic screening of 33,998 Chinese individuals revealed that over 90% of MMR gene variants were population-specific [13], emphasizing ethnic heterogeneity as a key source of inconsistency among studies.

In our cohort, exploratory analyses revealed differences in genotype distribution according to tumor stage, with a trend toward enrichment of the A allele in advanced disease. This is consistent with earlier findings in colorectal cancer showing preferential association of the −93G>A variant with MSI-positive tumors [28]. Similarly, Malik et al. demonstrated that reduced MLH1 expression in breast tumors correlates with higher grade and aggressive clinical features, supporting the hypothesis that attenuation of MLH1 function may influence not only tumor initiation but also progression [29]. In addition, no significant variation in genotype frequencies was observed across histological subtypes, with invasive ductal carcinoma predominating. Although the sample size for both lobular and rare subtypes was limited, these data provide initial insight and merit confirmation in larger, histologically balanced cohorts.

Most common breast cancer risk variants identified in genome-wide studies exert modest effects (OR = 1.1–1.3) [21]. The association we observed for *MLH1* −93G>A (OR ≈ 1.7) exceeds this range, which could reflect either population-specific effects or limited sample size rather than a true strong effect. While individual variants confer minor risk, their cumulative impact can be substantial when incorporated into polygenic risk models. Given that populations such as Azerbaijan are underrepresented in large GWAS, studies like ours provide valuable insights for building more globally inclusive risk prediction models.

Our study has certain limitations that should be taken into account when interpreting the results. The number of individuals with the AA genotype was relatively small, which reduced the precision of estimates in the recessive model and in subgroup analyses. In addition, we focused on a single polymorphism in *MLH1*; extending the analysis to other loci and haplotypes across the MMR pathway would provide a more comprehensive picture. Finally, although we evaluated several genetic models, these represent different interpretations of the same variant. After correcting for multiple comparisons using the false discovery rate, the observed associations no longer reached statistical significance. Taken together, these considerations suggest that our findings should be viewed as preliminary and hypothesis-generating, and they highlight the need for confirmation in larger, independent studies.

## 5. Conclusions

In conclusion, our findings suggest that *MLH1* −93G>A polymorphism may modestly contribute to BC susceptibility, with the dominant model showing an increased risk in our cohort. Although this association did not remain significant after correction and, therefore, requires validation, it aligns with reports from several Asian populations and with broader evidence linking *MLH1* promoter variation to cancer risk. Data on this variant in breast cancer remain very limited, and our study represents the first evidence from Azerbaijan. By documenting this association in a population largely absent from global genomic analyses, we contribute to expanding the international understanding of genetic diversity in breast cancer risk. The biological plausibility of this association, supported by the established role of *MLH1* in maintaining genomic stability, strengthens the relevance of our results. While the effect size is modest, it is in line with the contribution expected from common regulatory variants and may acquire predictive value when integrated into polygenic models of BC risk. Future studies with larger cohorts, broader MMR gene coverage, and functional assays such as promoter methylation and expression analyses will be critical to confirm these observations and clarify the precise role of *MLH1* −93G>A in breast carcinogenesis.

## Figures and Tables

**Figure 1 biomedicines-13-02769-f001:**
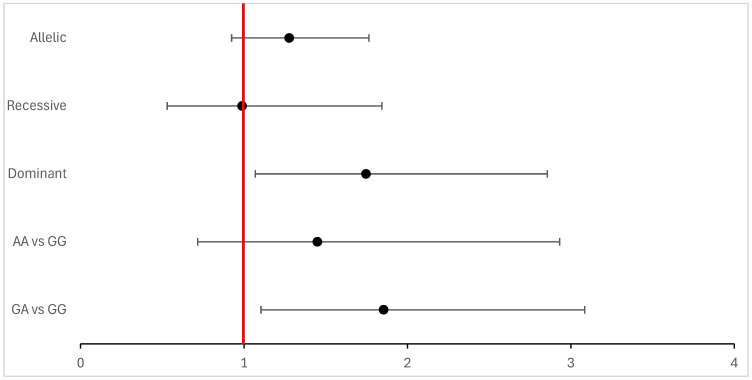
Forest plot showing odds ratios (ORs) and 95% confidence intervals for the association between MLH1 −93G>A polymorphism and breast cancer risk under co-dominant, dominant, recessive, and allelic models (reference = GG). The vertical red line represents the null value (OR = 1).

**Table 1 biomedicines-13-02769-t001:** Characteristics of study groups.

Clinical Characteristics	Patients, *n* = 143 (%)	Controls, *n* = 161 (%)	*p* Value
**Age** Age interval, mean	39–85 (62 ± 11)	32–83 (60 ± 11)	0.432
**BMI** ^1^ BMI (kg/m^2^) mean ± SD	28.1 ± 4.9	23.5 ± 3.8	**<0.001**
**Histological type**			
Invasive ductal carcinoma (IDC)	122 (85.3)		
Invasive lobular carcinoma (ILC)	18 (12.6)		
Other special types	3 (2.1)		
**Estrogen receptor (ER)**	Positive 96 (67.1)		
**Progesterone receptor (PR)**	Positive 91 (63.6)		
**HER2 status**	Positive 28 (19.5)		
**Triple-negative phenotype**	7 (4.9)		
**Tumor Grade**			
G1	26 (18.2)		
G2	103 (72.0)		
G3	14 (9.8)		
**Tumor Stage**			
I	14 (9.8)		
II	59 (41.2)		
III	14 (9.8)		
IV	56 (39.2)		
**CA 15-3** CA 15-3 (U/mL) mean ± SD	78.4 ± 52.6		
**BRCA1/2 status**	Negative 79 (55)		

^1^ Detailed BMI category, CA 15-3 and other clinical data are presented in Appendix A. Bold *p*-value denotes statistical significance.

**Table 2 biomedicines-13-02769-t002:** Distribution of *MLH1* –93G˃A polymorphism in patients and controls ^1^.

Genotype	Patients, *n* = 143 (%)	Controls, *n* = 161 (%)	OR (95% CI)	*p* Value
GG	37 (25.9)	61 (37.9)	Reference	-
GA	84 (58.7)	75 (46.6)	1.855 (1.104 –3.085)	0.019
AA	22 (15.4)	25 (15.5)	1.450 (0.717–2.932)	0.299
Dominant model				
GG	37 (25.9)	61 (37.9)	Reference	-
GA + AA	106 (74.1)	100 (62.1)	1.747 (1.069–2.856)	0.026
Recessive model				
GG + GA	121 (84.6)	136 (84.5)	Reference	-
AA	22 (15.4)	25 (15.5)	0.989 (0.530–1.844)	0.972
Allele				
G	158 (55.2)	197 (61.2)	Reference	-
A	128 (44.8)	125 (38.8)	1.277 (0.924–1.764)	0.138

^1^ Reference category: GG genotype. Odds ratios (ORs) with 95% confidence intervals (CIs) were calculated using logistic regression.

**Table 3 biomedicines-13-02769-t003:** Analysis of *MLH1−93G˃A* genotypes according to age in the studied group.

Category	Genotype	Patients *n* (%)	Controls *n* (%)	OR (95% CI)	*p* Value
**Age ≤ 60 years**; (Patients *n* = 67; Controls *n* = 107)	GG	18 (26.9)	41 (38.4)	Reference	-
	GA	42 (62.7)	49 (45.8)	1.948 (0.976–3.891)	0.050
	AA	7 (10.4)	17 (15.8)	0.942 (0.335–2.653)	0.903
**Age > 60**; (Patients *n* = 76; Controls *n* = 54)	GG	19 (25)	20 (37)	Reference	-
	GA	42 (55.2)	26 (48.2)	1.702 (0.767–3.772)	0.191
	AA	15 (19.8)	8 (14.8)	1.974 (0.683–5.713)	0.210

**Table 4 biomedicines-13-02769-t004:** Distribution of the *MLH1* −93G˃A genotypes in terms of tumor grade and stage.

Category	Subgroup	GG *n* (%)	GA *n* (%)	AA *n* (%)	*p* Value
Tumor grade	G1	10 (38.5)	15 (57.7)	1 (3.8)	0.047
	G2	23 (22.3)	64 (62.2)	16 (15.5)	
	G3	4 (28.6)	5 (35.7)	5 (35.7)	
Tumor stage	I	4 (28.6)	5 (35.7)	5 (35.7)	0.013
	II	14 (23.7)	33 (55.9)	12 (20.4)	
	III	3 (21.4)	7 (50)	4 (28.6)	
	IV	16 (28.6)	39 (69.5)	1 (1.9)	

## Data Availability

The original contributions presented in this study are included in the article and Appendix A. Further inquiries can be directed to the corresponding authors.

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
