# Peer review of "MLH1 Promoter Variant −93G>A and Breast Cancer Susceptibility: Evidence from Azerbaijan"

_biomedicines, 2025, doi:10.3390/biomedicines13112769_

Round 1

Reviewer 1 Report

Comments and Suggestions for Authors

The manuscript entitled “MLH1 Promoter Variant −93G>A and Breast Cancer Susceptibility: Evidence from Azerbaijan” by Karimova et al. investigates MLH1 -93G>A polymorphism in breast cancer patients from Azerbaijan. This a well-structured study which addresses an important gap in regional cancer genetics by establishing a novel population-specific association between MLH1 −93G>A and breast cancer risk. But several methodological and presentation issues need to be addressed before meeting the standards for publication in the journal. The areas of improvement are suggested below:

(1) The authors are advised to provide clearer justification for the chosen sample size. Additionally, the study would benefit from the additional details on the important variables like family history, BMI, menopausal status, hormone receptor status (ER/PR/HER2), BRCA status etc.

(2) For enhancement of the interpretability, the authors are requested to represent the data in bar chart or forest plots instead of table format. Kaplan-Meier curves can also be included if the survival data is available.

(3) The abstract is too dense and lengthy and full of complex sentences. The authors should restructure the abstract to focus on main finding. Overall, the sentences can be simplified, and grammatical refinement is also necessary. The authors should use some connector words like “while”, “however” to enhance the readability of the manuscript.

(4) Although FDR analysis has been done, the main findings lose its significance after FDR correction, and this should be clearly mentioned in the ‘Abstract’. The authors should be discussing about the biological significance of the study despite statistical limitations.

I believe addressing these comments would make the article suitable for publication in this journal.

Comments on the Quality of English Language

The abstract is too dense and lengthy and full of complex sentences. The authors should restructure the abstract to focus on main finding. Overall, the sentences can be simplified, and grammatical refinement is also necessary. The authors should use some connector words like “while”, “however” to enhance the readability of the manuscript.

Author Response

We sincerely thank the reviewer for the constructive and thoughtful comments on our manuscript.

We highly appreciate the reviewer’s recognition of the study’s contribution to regional cancer genetics and the valuable suggestions that have helped us substantially improve the manuscript’s clarity, methodological transparency, and overall quality.

Below, we address each point in detail.

Comment 1:  The authors are advised to provide clearer justification for the chosen sample size. Additionally, the study would benefit from the additional details on the important variables like family history, BMI, menopausal status, hormone receptor status (ER/PR/HER2), BRCA status etc.

Response 1:

We thank the reviewer for this important observation. We have now expanded the Materials and Methods section to clarify the sample size rationale and provide additional patient information. The study was designed as a pilot case–control study, and the number of participants (143 breast cancer cases and 161 healthy controls) was based on recruitment of the limited local cohort of ethnically Azerbaijani women. Furthermore, we have now included additional clinical variables as recommended: BMI values were calculated for all participants and presented in Table 1; BRCA mutation status is now specified, indicating that only sporadic, BRCA-negative cases were included to eliminate hereditary bias; Hormone receptor status (ER/PR/HER2) and histopathological data are now summarized to provide a more comprehensive clinical characterization of the patient cohort (lines 99-105, 159-174, Table 1, Table S1, S2, S3 (lines 493) (supplementary materials)).
We acknowledge that menopausal status was not available for all patients, thus, it was not included.

Comment 2: For enhancement of the interpretability, the authors are requested to represent the data in bar chart or forest plots instead of table format. Kaplan-Meier curves can also be included if the survival data is available.

Response 2:

We sincerely thank the reviewer for this valuable suggestion. We fully agree that a visual representation enhances interpretability. Accordingly, we have included a forest plot (Figure 1) summarizing the odds ratios (ORs) and 95 % confidence intervals for all genetic models (co-dominant, dominant, recessive, and allelic).

We considered bar charts; however, we found that forest plots provide a more accurate and statistically appropriate visualization for association studies of this type. For completeness, the detailed numeric data remain in Table 2, ensuring transparency and allowing exact values to be referenced.

Since survival data were not available for all  study subjects, Kaplan–Meier analysis could not be performed.

Comment 3:

The abstract is too dense and lengthy and full of complex sentences. The authors should restructure the abstract to focus on main finding. Overall, the sentences can be simplified, and grammatical refinement is also necessary. The authors should use some connector words like “while”, “however” to enhance the readability of the manuscript.

Response 3:

We are grateful for this valuable stylistic feedback. The abstract has been revised for clarity and conciseness. We have simplified the sentence structure and emphasized the main finding of the investigation (lines 12-30). Additionally, we reviewed the full manuscript for stylistic and grammatical refinement. Several long sentences in the Introduction and Discussion sections were shortened or restructured for smoother reading and consistency.

Comment 4:

Although FDR analysis has been done, the main findings lose its significance after FDR correction, and this should be clearly mentioned in the ‘Abstract’. The authors should be discussing about the biological significance of the study despite statistical limitations.

Response 4:

We thank the reviewer for this helpful comment. The loss of statistical significance after FDR correction was already mentioned in the Abstract (line 30). To further align with the reviewer’s suggestion, we have now added a concise statement emphasizing the biological significance of the MLH1 −93G>A promoter variant, highlighting its potential functional relevance to mismatch repair and breast cancer pathogenesis despite statistical limitations (lines 12-38).

Reviewer 2 Report

Comments and Suggestions for Authors

Review on the manuscript titled “MLH1 Promoter Variant −93G>A and Breast Cancer

Susceptibility: Evidence from Azerbaijan” by Karimova et al., 2025.

The authors addressed the task of evaluating MLH1 promoter variant association with Breast Cancer (BC) within the population of 143 women with histologically confirmed breast cancer against 161 individuals control without BC pathology targeted in the study.

As the authors convey in the introduction, the MLH1 -93G>A impact was surveyed previously in a range of studies (PMID: 29304767 (dysplastic sessile serrated adenomas); PMID: 18615680 (colon cancer)). The latest study (PMID: 21745804) surveyed the risk of the ‘A’ substitution in a large cohort (13 691 cancer cases and 14 068 controls from 17 published studies) revealed ‘borderline significance association between the A substitution and cancer risk. The authors also stress the ethnos specificity of the risk: the Asian population were more prone to cancer risk than Europeans. Also, the authors therein outlined that ‘ elevated cancer risks were observed in hospital-based studies but not in population-based studies. These findings showed no persuasive evidence that MLH1 -93 G/A polymorphism was associated with an increased risk of cancer’.

Thus, while there are instances of association of MLH1 mutation with cancer, there are no complete evidence as of today.

The authors genotyped both cohorts for MLH1 genotypes with two methods: PCR-RFLP and NGS. The results of analysis are presented in 4 tables:

Table 1. Characteristics of study groups: the authors observed quite close age distribution in both groups, though Tumor stage groups revealed rather large non-even variations across them with groups II and IV elevated, while tumor grades were normally distributed .

Table 2. Distribution of MLH1 –93G˃A polymorphism (genotypes?) in patients and controls1.

The authors considered dominant and recessive models while assessing the difference between groups, with some maintaining statistically significant difference.

Table 3. Analysis of MLH1-93G˃A genotypes (distribution) according to age in the studied group(s).

The table stratify the samples into gender groups for statistical significance assess.

We can see drastic downturn of the normal GG affected females <60yrs genotypes compared to the control, and some other specific observation based on the table.

Last Table: Table 4. “Distribution of the MLH1 -93G˃A genotypes in terms of tumor grade and stage” features significant difference between the groups in tumor grade, and most specifically, in tumor stage.

                In the consequent discussion, the authors present the main inferences from the study. There’s a hypothesis underlined in the paper’s title that, based on the previous assessments, the MLH1 effect is population-specific, and probably may prove relevant in Azeri population. It still would render a big multifactorial cohort to assess this effect with higher confidence as stated in the previous works.

Overall, the authors of the manuscript acknowledged in their study that they observe borderline statistical significance of MLH1(A) in the affected group comparing genotype distributions.

Formally, the manuscript is transparent, logically structured, and presents the results that may be of interest to the researchers/clinicians in the field, as well as used in further association studies.

Author Response

We sincerely thank the reviewer for their thoughtful and constructive evaluation of our manuscript. We truly appreciate the positive comments regarding the study’s structure, clarity, and potential relevance to the field of cancer genetics. We also thank the reviewer for recognizing our acknowledgment of the limited statistical power and population-specific scope of the study. We agree that broader, multi-center cohorts will be necessary to confirm these preliminary findings, and we have emphasized this point more explicitly in both the Discussion and Conclusions sections.

Reviewer 3 Report

Comments and Suggestions for Authors

The reviewed manuscript is focused on genotyping of MLH1, one of the genes that contributes to DNA mismatch repair and is therefore associated with an increased risk of carcinogenesis. The MLH1-93G>A (rs1800734) promoter polymorphism has been studied for at least two decades in patients with various malignancies - colorectal cancer, lung, ovary, prostate, bladder, and brain. Research on breast cancer is limited and its results are ambivalent which makes the reviewed study relevant.

It should be noted that this is a local study performed in Azerbaijan in which only women of Azerbaijani origin were included. The restriction in patient selection is based on the results of earlier studies which showed that the MLH1-93G>A polymorphism significantly depends on the ethnicity of a patient.

This study is a pilot one. It includes 143 breast cancer patients and 161 healthy donors. The statistical focus of the study makes this number of patients insufficient for appropriate conclusions. However, the authors acknowledge that they have provided "food for thought" which still needs to be confirmed by further studies involving a significantly larger number of patients and a more diverse ethnic background.

The article's abstract provides a comprehensive overview of the study and its conclusions.

The Introduction section presents data on MLH1 and highlights its central role in DNA mismatch repair which is associated with various consequences of its functional activity and clinical disorders.

Since MLH1 polymorphism, its impact on DNA mismatch repair and clinical associations with this gene have been studied since the beginning of this century, the authors include a significant number of early studies in the reference list, with publications of the last 5 years accounting for approximately 30%. It is advisable to expand the reference list by including studies performed in recent years and ongoing now (if they have been published online).

The "Materials and Methods" section contains a brief description of the subjects included in the study: patients and healthy donors. It is more common in scientific literature to put a detailed description of the patients into this section, primarily, the disease characteristics, but the authors include these parameters in the "Results" section. Table 1 presents the patient stratification by age, stage, and tumor grade. This is acceptable, but still it would be more appropriate to move Table 1 to the "Materials and Methods" section. As it is, Table 1 contains only patient characteristics without any results obtained in the study.

A serious disadvantage, in our opinion, is lack of information regarding the histological type of breast cancer diagnosed in the involved patients. It is well-known that it is not correct to speak about breast malignancy without mentioning its histological form and subtype. Each tumor type and subtype has its own characteristics in terms of growth rate, metastasis intensity, and patient survival. It is likely that MLH1 polymorphism may be closely associated with the characteristics of each type of the disease. For example, stage IV of the disease might have been predominantly present in the triple-negative cancer patients, and this could be the reason for the observed difference in the allele distribution. Further analysis and discussion of this issue is needed.

So, we should support the authors' conclusion about a pilot character of the study. The definitive conclusions about the role of MLH1 polymorphism in breast cancer can be made only after a study involving a significantly larger cohort of patients and validation of the genotyping method, with regard to the histological type and subtype of breast cancer.

This manuscript requires significant revision and re-review, and cannot be recommended for publication in its current form.

Author Response

We sincerely thank the reviewer for the thoughtful and detailed assessment of our manuscript and for acknowledging its relevance, the quality of the abstract, and the pilot character of the study. We have carefully revised the manuscript in response to all comments. The corresponding modifications are outlined below.

Comment 1:

Since MLH1 polymorphism, its impact on DNA mismatch repair and clinical associations with this gene have been studied since the beginning of this century, the authors include a significant number of early studies in the reference list, with publications of the last 5 years accounting for approximately 30%. It is advisable to expand the reference list by including studies performed in recent years and ongoing now (if they have been published online).

Response 1:

We fully agree and have updated the Introduction and Discussion with three new recent references (2023–2025) reflecting current literature on MMR biology and breast-cancer genetics:

  1. Xiong et al., 2025 (Signal Transduct. Target. Ther. 10, 49) – overview of hereditary vs. sporadic breast cancer mechanisms.
  2. Kozonoe et al., 2024 (Surg. Exp. Pathol. 7, 13) – recent evidence of MMR deficiency in breast cancer.
  3. Mbuya-Bienge et al., 2023 (Cancers 15, 5380) – polygenic risk and low-penetrance variants in breast-cancer susceptibility.

These additions increased the proportion of references from the last five years from appx. 30 % to ≈40 %.

Comment 2 and 3:

 The "Materials and Methods" section contains a brief description of the subjects included in the study: patients and healthy donors. It is more common in scientific literature to put a detailed description of the patients into this section, primarily, the disease characteristics, but the authors include these parameters in the "Results" section. Table 1 presents the patient stratification by age, stage, and tumor grade. This is acceptable, but still it would be more appropriate to move Table 1 to the "Materials and Methods" section. As it is, Table 1 contains only patient characteristics without any results obtained in the study. A serious disadvantage, in our opinion, is lack of information regarding the histological type of breast cancer diagnosed in the involved patients. It is well-known that it is not correct to speak about breast malignancy without mentioning its histological form and subtype. Each tumor type and subtype has its own characteristics in terms of growth rate, metastasis intensity, and patient survival. It is likely that MLH1 polymorphism may be closely associated with the characteristics of each type of the disease. For example, stage IV of the disease might have been predominantly present in the triple-negative cancer patients, and this could be the reason for the observed difference in the allele distribution. Further analysis and discussion of this issue is needed.

Response 2 and 3:

We thank the reviewer for this valuable and constructive comment. In the revised version, in order to provide a more comprehensive overview of the studied cohort, we have significantly expanded the clinical dataset to include BMI, histological types, receptor status (ER, PR, HER2, TNBC), BRCA1/2 status, TNM classification and even CA 15-3 levels (lines 99-105, 159-174, Table 1, line 493).

While we appreciate the reviewer’s suggestion to relocate Table 1, we have decided to retain the table within the Results section. This decision was made because the table now contains both descriptive characteristics and analytical data and keeping it in the Results section ensures a logical flow between the narrative and the presented data.

To maintain focus and clarity, we have placed extended supporting data of histological subtypes with MLH1 −93G>A genotypes (Supplementary Table S3) and detailed clinicopathological parameters (Supplementary Table S1, S2). A brief reference to these analyses has been added to the Discussion to acknowledge the exploratory nature of these findings while avoiding over-interpretation due to limited statistical power (lines 285-289).

Round 2

Reviewer 1 Report

Comments and Suggestions for Authors

The authors have successfully addressed all the comments. Inclusion of the BMI and hormone receptor statuses and the Forest plot definitely enhanced the quality of the manuscript. Writing has also been improved. I think the manuscript is now suitable for publication in the journal.

Reviewer 3 Report

Comments and Suggestions for Authors

The manuscript is revised and may be recommended for publication in the Biomedicines in the present form.